# Intelligent Power Unit Parameters Design and the Influence Analyses

**Zhanshan Zhu \*, Huichao Zhao, Xiaolu Liu, Hongbao Wang and Zhiqiang Liu**

E-Drive System Department-Faw, Changchun 130000, China; zhaohuichao@faw.com.cn (H.Z.); liuxiaolu@faw.com.cn (X.L.); wanghongbao@faw.com.cn (H.W.); liuzhiqiang@faw.com.cn (Z.L.)

\* Correspondence: zhuzhanshan@faw.com.cn; Tel.: +86-187-4401-8277

**Abstract:** The power unit is mainly responsible for the power converter's function of the electric vehicle, which converts the DC power from the battery to the AC power to drive the motor. Therefore, the parasitic parameters of the power unit will directly affect the output performance of the vehicle. In this paper, the parasitic parameters of the power unit are analyzed. By using an asymmetric design, the capacitor cost and performance are balanced. Moreover, the structure of the busbar is optimized according to the reasonable theoretical analysis. The simulation comparison before and after the optimization, and the comparison results of the measurement results by the double pulse test, are given to verify the optimization. Additionally, the ESL of the busbar is reduced by 10 nH. The results will offer some references to the power unit design.

**Keywords:** power unit; simulation optimization; surge peaks; equivalent series inductance

## 1. Introduction

With the limitation of carbon emissions and the popularity of renewable energy vehicles, more and more consumers choose to buy or drive renewable energy vehicles. As the core part of the power system of renewable energy vehicles, the performance of the power unit will directly affect the acceleration performance and energy consumption of the vehicle. The performance of the power unit is strongly related not only to the characteristics of the components it chooses, but also to the parasitic parameters of each part in the power unit.

The parasitic parameter distribution of the whole power unit is analyzed in this paper. The power unit is mainly divided into the following three parts: capacitor, busbar, and power module [1]. After the power module is selected, its parasitic parameter is determined, so this will not be discussed in detail. This paper will discuss the influence of the capacitor shape on the parasitic parameters. An analysis and simulation of the busbar shape on the parasitic parameters will also be given.

## 2. Capacitor Parasitic Parameter

### 2.1. Equivalent Series Resistance of Capacitor

The DC-link capacitor plays the role of short-term energy storage and peak filling in the inverter. The DC-link capacitor has three key electrical parameters, capacitance (C), equivalent series resistance (ESR), and equivalent series inductance (ESL). The capacitance mainly affects the ripple voltage of the capacitor. The capacitance is usually positively related to the volume of the capacitor. The larger the volume, the larger the capacitance. The ESR of capacitors mainly affects the thermal performance of capacitors. Usually, the thermal performance of capacitors is the main factor that limits the continuous performance of inverters. The smaller the ESR, the lower the temperature rise of the capacitors during operation. A lower temperature rise can prolong the life of the capacitor [2]. Most capacitors are made up of several single capacitor cores in parallel. There are three main factors that

affect the ESR of a capacitor single core, which can also be equivalent to the resistance formula, such as Formula (1) and Figure 1.

$$R = \rho \frac{L}{S}$$

(1)

$R$: capacitor equivalent resistance (ESR);
$L$: a single core length;
$S$: a single core cross-sectional area;
$\rho$: metallized film equivalent resistivity.

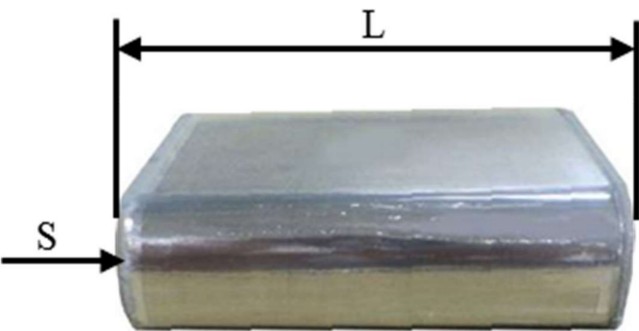

**Figure 1.** Schematic diagram of a capacitor core.

The longer the core, the larger the ESR. The larger the cross-sectional area, the smaller the ESR. The metallized film equivalent resistivity relates to the evaporation process and safety grid style.

Because the temperature resistance of the polypropylene film in the capacitor is lower than that of other parts inside the inverter, it is a very effective method to reduce the ESR of the capacitor by rationally designing the shape of the capacitor according to Formula (1), to reduce the heat generation of the capacitor itself. Therefore, the shape of the capacitor is as low as possible, similar to a pie shape in Figure 2. In this way, the resistance of the capacitor itself is low and the heat generation will be small. Meanwhile, the surface area of heat dissipation is larger than that of the cube shape, which is beneficial to the continuous performance of the capacitor, and the thermal resistance is also small. However, the pie shape of the capacitor means that some capacitor cores are far away from the terminals connected with IGBT. With the influence of the current skin effect at high frequency, the capacitor's core far away from IGBT will not be effective. When the inverter uses a silicon carbide device and operates at high frequency [2], the design of the shape of a capacitor should consider both thermal and electrical performance.

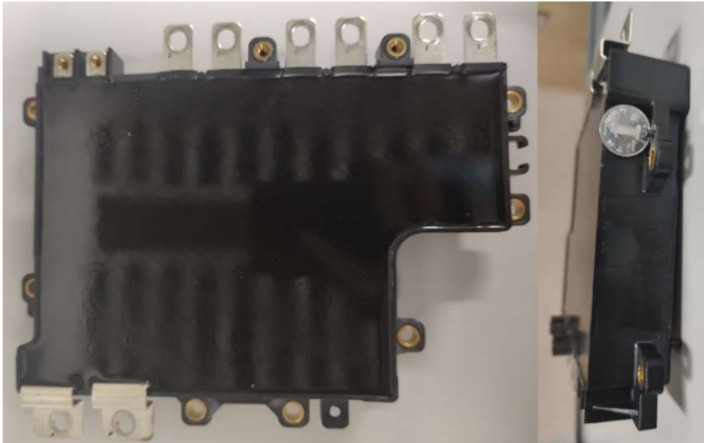

**Figure 2.** A low-type capacitor.

Although the ESR of the capacitor should be reduced as much as possible, a lower ESR usually requires a larger capacitance and increases the number of core capacitors, or a thicker positive and negative busbar to reduce parasitic resistance from the busbar. The combination of the two methods will increase the cost of the capacitor. Some capacitors have cooling water flowing through the bottom plate [3]. Copper has high thermal conductivity. When the bottom plate of the capacitor has a good cooling condition, the positive and negative busbars can be used for asymmetric design. The capacitor with cooling adopts a thinner busbar, and the capacitor without cooling adopts a thicker busbar, so that the cost and performance will be balanced. We designed two capacitors with the shape shown in Figure 3, where only the busbars with cooling are different. At the same inverter test condition (environment temperature: 85 °C, cooling temperature: 65 °C) and the same motor load with SVPWM control (Vdc: 650 V, phase current: 180 Arms), the two capacitors' thermal simulations and tests only have a difference of 2 °C; the capacitor with a thinner busbar temperature is higher.

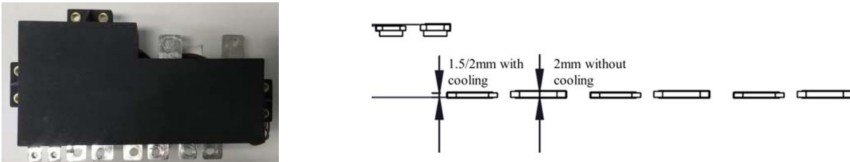

**Figure 3.** A capacitor with a different busbar.

### 2.2. Equivalent Series Inductance of Capacitor

The ESL of the capacitor mainly affects the surge peak of IGBT [4,5]. At the same time, IGBT also has ESL. Generally, the larger the ESL of the whole circuit, the higher the surge peaks when IGBT switches [6]. Similarly to ESR, the value of the ESL of the capacitor is also strongly related to the shape of the capacitor. The shape of a lower ESL capacitor should have the following characteristics: (1) the power terminals should be as short as possible, (2) the power terminal and the internal busbar of the capacitor should be laminated with the maximum area such as Figure 4, and (3) the terminal should be located in the middle of the whole capacitor to avoid the group terminals' ESL being larger than others, which will become the weak point of the design.

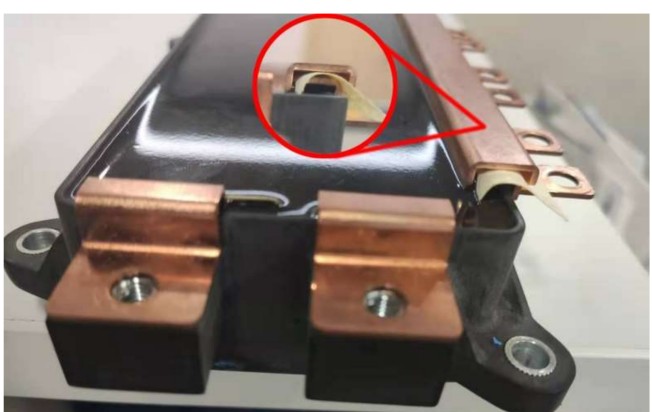

**Figure 4.** Capacitor with laminated busbars.

### 3. Equivalent Parameters of Busbar

Busbars are electrical connections in power units, so their parasitic parameters will directly affect the characteristics of the power unit. The parasitic parameters of the busbar are mainly divided into the equivalent series resistance (ESR) and the equivalent series inductance (ESL). Additionally, in engineering applications, the ESL of the busbar is

harder to design, and the inductance of the long round wire can be simply estimated by Formula (2), as follows:

$$L = \frac{\mu_0 l}{2\pi}\left(\ln \frac{2l}{r} - 0.75\right) \tag{2}$$

$L$: inductance of circular section straight wire;
$l$: wire length;
$r$: conductor radius;
$\mu_0$: vacuum permeability.

It can be observed from Formula (2) that the inductance is related to the conductor length and cross-sectional area. When the currents of two busbars are opposite and laminated, the magnetic fields generated by the currents in the opposite direction cancel each other out, reducing the magnetic field energy stored in the air around the busbar. The principle can be characterized by the ampere circuit law and Formula (3).

$$\oint \vec{B} \cdot \vec{dl} = \mu_0 \sum_{i=1}^{n} I \tag{3}$$

$B$: magnetic flux density;
$I$: current in the loop;
$dl$: the loop differential;
$\mu_0$: vacuum permeability.

Because the current flowing through the two busbars is equal and reverse, the loop integral of the current is zero. In fact, the two busbars are not completely coupled, so the magnetic field will be generated around the busbar.

By increasing the laminated area between the two busbars, reducing the gap between the two busbars, or adjusting the laminated structure of the busbars, the coupling degree of the two busbars can be effectively improved. The inductance of the whole loop will be reduced.

### 3.1. Busbar Simulation and Optimization

The inductance of two long round wires could be simply estimated by Formula (4). Although the bus cross-section is not round, it has a similar structure. We can use Formula (4) as a guide to reduce the busbar inductance by reducing $l$ and D.

$$L = \frac{\mu_0 l}{\pi}\left(\ln \frac{D}{R} + 0.25\right) \tag{4}$$

$L$: two wires' self-induction;
$l$: wire length;
D: the distance between the two wires;
$R$: conductor radius.

In order to verify the optimization effect of a laminated busbar on ESL, this paper simulates the ESL of the following four different busbars: (A) common shape; (B) adding laminated area; (C) three-layer composite structure (two positive busbars and one negative busbar); and (D) three-layer and edge welding structure. According to the Formulas (3) and (4), the ESL will be lower and lower. Then, we used 3D-FEM software to calculate the four designs of the ESL. The A–D busbar shapes are shown in Figure 5. The simulation result at 10 kHz is shown in Table 1.

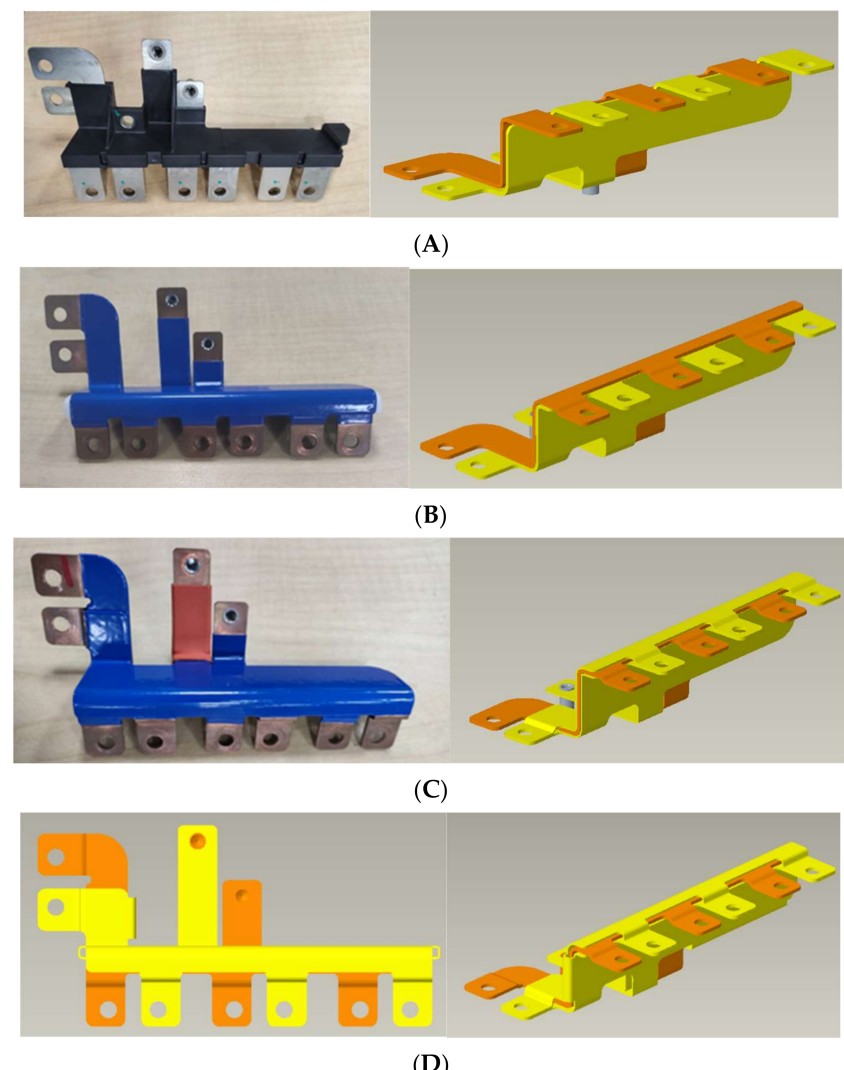

**Figure 5.** Busbar outline drawing. (**A**) Common shape; (**B**) adding laminated area; (**C**) three-layer composite structure; (**D**) three-layer and edge welding structure.

**Table 1.** ESL simulation value of four busbars.

| Busbar | ESL (Simulation) @10 kHz |
|---|---|
| A | 42 nH |
| B | 29 nH |
| C | 27 nH |
| D | 23 nH |

### 3.2. Busbar Measurement

To verify the effect of busbar optimization, a double pulse test of the power unit under the same working condition is tested. The double pulse test waveform of the whole power unit is shown in Figure 6, and the four waveform scales are as follows: yellow 5 V/div, blue 10 V/div, green 200 V/div, and red 200 A/div (we used a current probe, so the unit is A/div). It can be observed that the surge voltage is high between the collector and emitter of IGBT due to the existence of inductance in the whole power circuit during IGBT shutdown. Moreover, the ESL of the busbar could be calculated by Formula (5). $L_{module}$ and $L_{cap}$ are already known as 9.1 nH and 14 nH, respectively, in their datasheet.

$$V_{ce} = V_{dc} + (L_{busbar} + L_{module} + L_{cap}) \times \frac{di}{dt} \tag{5}$$

$V_{ce}$: the maximum voltage at IGBT power terminal when IGBT turns off;
$V_{dc}$: the test condition DC voltage;
$L_{module}$: the ESL of the power module;
$L_{cap}$: the ESL of the capacitor;
$di/dt$: the $I_c$ current rate.

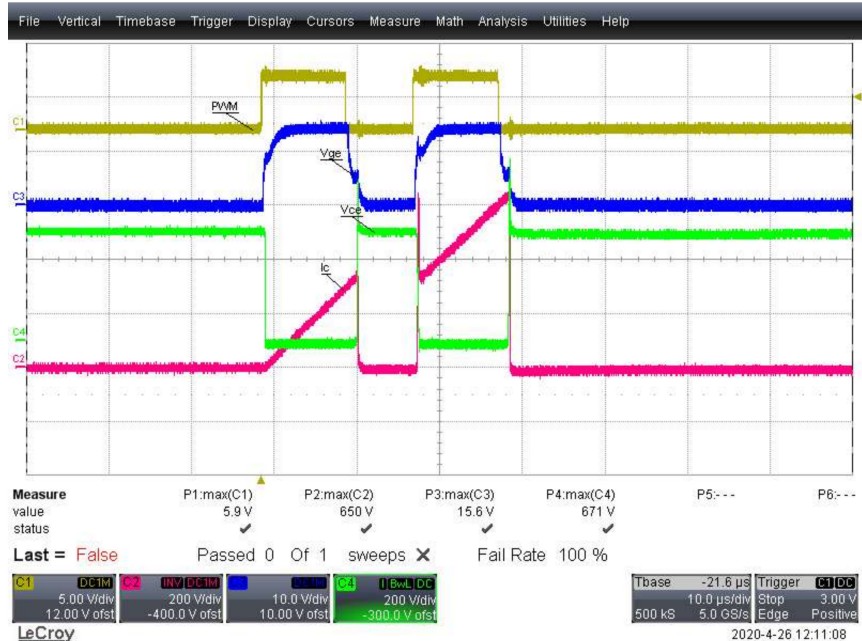

**Figure 6.** IGBT double pulse test waveform.

The surge voltage of three busbars under the same conditions is shown in Table 2. The test conditions are a bus voltage of 403 V, maximum current of 636 A, 23 °C, and Changchun atmospheric pressure.

**Table 2.** Maximum voltage at IGBT terminal.

| Busbar | Maximum Voltage at IGBT Power Terminal (V) | ESL (Calculation) (nH) |
|--------|---------------------------------------------|------------------------|
| A | 710 | 40 |
| B | 692 | 32 |
| C | 679 | 30 |

The measured results are consistent with the theoretical analysis and simulation analysis. By optimizing the laminated area and adjusting the laminated structure, the ESL of the busbar can be effectively reduced, to reduce the surge peak caused by IGBT switching. With a small surge peak, the inverter could operate at a higher voltage battery.

## 4. Conclusions

In this paper, the influence of the parasitic parameters of the capacitor are analyzed by combining them with the structure shape of the capacitor, and design suggestions are given. At the same time, the theory of the busbar ESL is briefly explained. The simulation and double pulse measurement verify that by increasing the laminated area and adjusting the laminated structure between three shapes, the proposed design can effectively reduce the ESL of the busbar by 20%, while keeping the shape of the busbar similar. A lower ESL can reduce the IGBT switching surge peak. By analyzing the parasitic parameters of the key components in the power unit, the design of the power unit can be guided.

**Author Contributions:** Conceptualization, Z.Z.; Data curation, H.W.; Investigation, X.L.; Writing—review & editing, H.Z. and Z.L. All authors have read and agreed to the published version of the manuscript.

**Funding:** This research received no external funding.

**Institutional Review Board Statement:** Not applicable.

**Informed Consent Statement:** Not applicable.

**Data Availability Statement:** Not applicable.

**Conflicts of Interest:** All authors are employee of FAW Group. The paper reflects the views of the scientists, and not the company.

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
