# Peer review of "Intelligent Power Unit Parameters Design and the Influence Analyses"

_wevj, doi:10.3390/wevj12040238_

Round 1

Reviewer 1 Report

1. The article is concise and easy to read.
2. The comparison results of the four different combinations are very interesting, and can help to gain more design information from them.
3. It should be better if we can further discuss the influence of the combination method.

Author Response

Dear Reviewer:

    Thanks very much,the 3rd comment  is mentioned in the double pulse test already. because the waveforms are nearly the same(only the Vce max is different),so I only put one figure(fig.6) in the paper,and the results are showed in Tab.2.

Reviewer 2 Report

Below are notes to the revised article:

  1. Abstract written generally too, does not refer to the most important result obtained in this article.
  2. In the section "Introduction" - it describes what is to be done in the article, not the subject of the analysis. No reference to literature.
  3. In the section „Capacitor Parasitic Parameter” the information is generally known in the world of techniques, this information is not relevant to the literature. Moreover, the equation 1 shown is a basic formulation in the field of capacitors. There is a lot to do in this field, even in terms of capacitor excitation and electromagnetic inertia, the possibility of taking over large currents. There is such a sentence ... “We designed two capacitors, 79 which only the busbars with cooling are different. The two capacitors’ thermal 80 simulations and tests just have 2℃ different.” what assumptions for the project, nothing is known - what material, what structure, what gaps, etc. A sentence can be formatted to a different style by scoring. “The shape of a 88 lower ESL capacitor should have the following characteristics: (1) The power terminals 89 should be as short as possible, (2) The power terminal and the internal busbar of the 90 capacitor should be laminated as much as possible.(3) The terminal is located in the 91 middle of the whole capacitor to avoid on group terminals’ ESL larger than others, that 92 will become the weak point of the design.”
  1. The second equation relates to the circular sections of the wire - how does it relate to the designed and rectangular cross-sections? See figure 5.
  2. Figure 6 shows the simulation results but is unreadable. No axis description, legend etc. Is a screenshot. It is unacceptable at this level. What program on what assumptions was the simulation made.
  3. Where did the results in table 2 come from?
  4. Note: after reading the article, another scientist cannot repeat the simulation.
  5. The style of the article and the language used require correction, ect figures are not referenced in the text.
  6. Please refer to the literature. There is a lot of work on this topic in Energies or WEVJ, no reference was made to any of the experimental work.
  7. Conclusions too general and do not correspond to the title of the thesis. Where are the optimization issues described in the executive abstract.

Author Response

Dear Reviewer:

    Thanks very much,please see the attachment, I have updated my manuscript.

Reviewer 3 Report

  1. In the article the authors use the term optimization - no optimization procedure was found in the article. The use of the term optimization must result from a mathematical optimization procedure. The use of this word misleads the readers, it should be replaced by another term adequate to the analyses performed.
  2. On page 6 the term "theoretical analysis" is used - theoretical analysis was not found in the article.
  3. There is no description of what the quantities in Formulas 2 and 3 mean. 

Author Response

Dear Reviewer:

    Thanks very much,please see the attachment, I have update my manuscript.

Round 2

Reviewer 2 Report

Overall, more comments:

Poor literature review.

Figure 6 is hardly readable, this is a screenshot.

Abstract and summary, corrected does not refer to the most important information from the article. This will increase the readability of the article.

Author Response

Comments:Figure 6 is hardly readable, this is a screenshot.
Response:This is the original data,I think it is the best way to show the result.All the information can read from this figure

Reviewer 3 Report

The revised article still lacks a description of the optimization procedure. 

Author Response

Comments: The revised article still lacks a description of the optimization procedure.. Response:The optimization process can be seen in Figure 5 and Table 1, and the ESL from A to D is smaller and smaller.